# Multiple Mutations in the Non-Ordered Red Ω-Loop Enhance the Membrane-Permeabilizing and Peroxidase-like Activity of Cytochrome *c*

**DOI:** 10.3390/biom12050665

**Published:** 2022-05-04

**Authors:** Rita V. Chertkova, Alexander M. Firsov, Nadezda A. Brazhe, Evelina I. Nikelshparg, Zhanna V. Bochkova, Tatyana V. Bryantseva, Marina A. Semenova, Adil A. Baizhumanov, Elena A. Kotova, Mikhail P. Kirpichnikov, Georgy V. Maksimov, Yuriy N. Antonenko, Dmitry A. Dolgikh

**Affiliations:** 1Shemyakin-Ovchinnikov Institute of Bioorganic Chemistry, Russian Academy of Sciences, 117997 Moscow, Russia; zh.bo4kova@yandex.ru (Z.V.B.); tato-tato@list.ru (T.V.B.); marinaapbch@mail.ru (M.A.S.); kirpichnikov@inbox.ru (M.P.K.); dolgikh@nmr.ru (D.A.D.); 2Belozersky Institute of Physico-Chemical Biology, Lomonosov Moscow State University, 119991 Moscow, Russia; amfamf@yandex.ru (A.M.F.); kotova@genebee.msu.ru (E.A.K.); antonen@belozersky.msu.ru (Y.N.A.); 3Biophysics Department, Biological Faculty, Lomonosov Moscow State University, 119234 Moscow, Russia; evelinanikel@gmail.com (E.I.N.); adilbayzhumanov@me.com (A.A.B.); gmaksimov@mail.ru (G.V.M.); 4Biology Department, Lomonosov Moscow State University, 119899 Moscow, Russia; 5Federal State Autonomous Educational Institution of Higher Education “National Research Technological University “MISiS””, 119049 Moscow, Russia

**Keywords:** mitochondrial cytochrome *c*, heme, red Ω-loop of cytochrome *c*, liposome leakage, cardiolipin, peroxidase activity, resonance Raman spectroscopy, surface-enhanced Raman spectroscopy

## Abstract

A key event in the cytochrome *c*-dependent apoptotic pathway is the permeabilization of the outer mitochondrial membrane, resulting in the release of various apoptogenic factors, including cytochrome *c*, into the cytosol. It is believed that the permeabilization of the outer mitochondrial membrane can be induced by the peroxidase activity of cytochrome *c* in a complex with cardiolipin. Using a number of mutant variants of cytochrome *c*, we showed that both substitutions of Lys residues from the universal binding site for oppositely charged Glu residues and mutations leading to a decrease in the conformational mobility of the red Ω-loop in almost all cases did not affect the ability of cytochrome *c* to bind to cardiolipin. At the same time, the peroxidase activity of all mutant variants in a complex with cardiolipin was three to five times higher than that of the wild type. A pronounced increase in the ability to permeabilize the lipid membrane in the presence of hydrogen peroxide, as measured by calcein leakage from liposomes, was observed only in the case of four substitutions in the red Ω-loop (M4 mutant). According to resonance and surface-enhanced Raman spectroscopy, the mutations caused significant changes in the heme of oxidized cytochrome *c* molecules resulting in an increased probability of the plane heme conformation and the enhancement of the rigidity of the protein surrounding the heme. The binding of wild-type and mutant forms of oxidized cytochrome *c* to cardiolipin-containing liposomes caused the disordering of the acyl lipid chains that was more pronounced for the M4 mutant. Our findings indicate that the Ω-loop is important for the pore formation in cardiolipin-containing membranes.

## 1. Introduction

During the activation of the cascade of events leading to cell apoptosis, cytochrome *c* (CytC) moves across the outer mitochondrial membrane into the cytosol, where it either enhances the external apoptotic signal or initiates the activation of the caspase cascade by its own (CytC-dependent) apoptotic pathway [1,2]. Apoptotic processes are initiated by a variety of stimuli, including exposure to pathogens, radiation, chemotherapy drugs, and oxidative stress [3,4,5]. The key event in the CytC-dependent apoptotic pathway is the permeabilization of the outer mitochondrial membrane, as a result of which various apoptogenic factors, including CytC, enter the cytosol. It is generally accepted that this stage, controlled by the Bcl-2 proteins family, is a kind of bifurcation point for a cell (life or death) [3,4,5].

To date, it is believed that permeabilization of the outer mitochondrial membrane can be induced by the peroxidase activity of CytC, in turn induced by the formation of a complex between CytC and cardiolipin (CL) [6]. CL is a negatively charged phospholipid contained in the inner membrane of mitochondria that forms a membrane site for CytC binding. Upon the induction of apoptosis, a chain of events is triggered, leading to the movement of CL from the inner to the outer mitochondrial membrane, where a complex with CytC is formed [7,8,9]. In this case, significant conformational and functional rearrangements occur in the CytC molecule, as a result of which the protein loses its ability to transfer an electron and acquires a high peroxidase-like activity [10], causing the permeabilization of the outer membrane. At present, there is evidence in favor of pore formation occurring during the permeabilization of CL-containing membranes induced by the CytC peroxidase-like activity. In particular, in model systems of giant liposomes and planar bilayer lipid membranes, it was shown that CytC is capable of forming pores in a lipid membrane containing CL, both in the presence [11,12] and in the absence of hydrogen peroxide [13].

In this regard, it seems important to study the role of individual functionally significant amino acid residues or sequences of CytC in the interaction of CytC with CL-containing membranes. The data obtained will make a certain contribution to the determination of the molecular mechanisms of CytC interaction with mitochondrial membranes during its translocation into the cytosol, expanding our understanding of the role of conformational changes in CytC during the initial stages of apoptosis.

In previous works, we studied the interaction of CytC with natural and artificial lipid membranes [14], as well as the role of the Lys72 residue, which is essential for the functioning of protein, in the membrane-permeabilizing activity of CytC [15]. Previously, we have constructed a number of mutant CytC variants with several substitutions in the red Ω-loop 70–85 [16,17] and with multiple substitutions of surface Lys residues from the universal CytC binding site with its redox partners of the electron transport chain—ubiquinol–cytochrome *c*-reductase (complex III) and cytochrome *c*-oxidase (complex IV) [18]. In the course of a detailed study of these mutant variants, we found that the electron transport function of CytC with substitutions both in the universal binding site (CytC variant K8E/K27E/K72E/K86E/K87E/E62K/E69K/E90K (8Mut)) and in the non-ordered red Ω-loop (variants T78N/K79Y/M80I/I81M/F82N (M1), T78S/K79P (M2), I81Y/A83Y/G84N (M3) and P76I/G77L/I81L/F82L (M4)) is almost completely suppressed [16,18]. At the same time, in CytC mutants M1, M2, and M3, changes in the conformation and geometry of hemoporphyrin were observed, while the structure of the protein part of CytC was generally preserved [16,17].

In this work, we investigated the ability of the above CytC mutant forms to bind to cardiolipin, and their membrane-permeabilization and peroxidase activities. Using resonance and surface-enhanced Raman spectroscopy (RRS and SERS, respectively), we studied the conformation of reduced and oxidized heme molecules in wild-type (WT) CytC and two mutants with the most pronounced decrease in respiratory activity—K8E/K27E/K72E/K86E/K87E/E62K/E69K/E90K (8Mut) and P76I/G77L/I81L/F82L (M4). We also investigated conformational changes in heme and heme local surroundings in oxidized M4 and 8Mut CytC forms under their interaction with CL-containing liposomes. We also analyzed the influence of WT, M4, and 8Mut cytochromes on the ordering of lipid chains in CL-containing liposomes upon the interaction with CytC molecules.

## 2. Materials and Methods

Components for the culture media and buffer solutions for chromatography and electrophoresis (AppliChem, Darmstadt, Germany), ampicillin, CytC from horse heart (Sigma, Schnelldorf, Germany), *Xho* I restriction endonuclease (Promega, Madison, WI, USA), *Bam*H I restriction endonuclease (New England Biolabs Inc., Ipswich, MA, USA), *Pfu*-DNA polymerase, and T4-DNA ligase (Fermentas, Pabrade, Lithuania) were used in this study. Distilled water was additionally purified on a Milli-Q system (Millipore, Burlington, MA, USA). We used sodium dithionite (SDT) for the CytC reduction, bovine heart cardiolipin (in powder), soybean phosphatidylcholine, calcein, hydrogen peroxide, Triton X-100 (Sigma, Schnelldorf, Germany), luminol (MP Biomedicals, Eschwege, Hessen, Germany).

### 2.1. Construction of the Mutant Genes of CytC

The construction of amino acid substitutions in the CytC Ω-loop (70–85) sequence and the calculation of the structure of its mutant variants with substitutions T78N/K79Y/M80I/I81M/F82N (M1), T78S/K79P (M2), I81Y/A83Y/G84N (M3), and P76I/G77L/I81L/F82L (M4) were described earlier [16,17]. The construction of amino acid substitutions in the universal CytC binding site, during which the mutant variant K8E/K27E/K72E/K86E/K87E/E62K/E69K/E90K (8Mut) with altered surface charges was obtained, was described earlier [18].

The mutations were introduced into the gene of horse CytC in a composition with pBP(CYC1) expression plasmid vector by site-specific mutagenesis according to the QuikChangeTM Mutagenesis Kit method (Stratagene, La Jolla, CA, USA). The production of mutant DNA during mutagenesis was analyzed by electrophoresis in 1% agarose gel. The nucleotide sequences of mutant genes in the plasmid DNA were determined on an ABI Prism 3100-Avant Genetic Analyzer (Applied Biosystems, Beverly, MA, USA). The selected mutant genes were cloned in the pBP(CYC1) expression vector modified for the expression of genes of horse CytC [19].

### 2.2. Expression of the Mutant Genes of CytC, Protein Isolation, and Purification

The expression of the mutant genes of cytochrome c was performed in the JM-109 strain of *E.coli* in an SB liquid-nutrient medium with ampicillin (the final concentration was 200 µg/mL) without the addition of the inductor at 37 °C under vigorous stirring for 22–24 h [20].

Afterward, the growth cells were homogenized by forcing through a French press (Spectronic Instruments, Inc., Irvine, CA, USA) at high pressure with subsequent centrifugation at 100,000 g for 20 min.

The isolation and purification of the target proteins were performed on an “АKTA FPLC” liquid chromatographic system (GE-HEALTHCARE, CШA) according to the previously elaborated two-step scheme [16,18,21]. The degree of purification and concentration of cytochrome *c* in the resulting fractions were determined on a spectrophotometer and by SDS-PAGE electrophoresis. The fractions with the A_409_/A_280_ purity of 4.5–5.0 (this value corresponded to a purity of ≥95% for the substance commercially prepared by Sigma, Schnelldorf, Germany) were oxidized by treating with potassium ferricyanide added at the equimolar concentration, dialyzed three times against 10 mM ammonium carbonate buffer (pH 7.9), and lyophilized on an ALPHA I-5 device.

### 2.3. Centrifugation Binding Assay

Protein-liposome ultracentrifugation binding assays were performed as previously described [22,23,24], with certain modifications. In particular, liposomes were prepared in the buffer containing 200 mM sucrose and subsequently dispersed in the buffer containing 200 mM glucose. The measurement solution contained 180 mM glucose, 10 mM KCl, 10 mM Tris, 10 mM MES, 1 mM EDTA, pH 7.4; the lipid concentration was 50 mg/mL. CytC (WT or mutant variants) was added at a concentration of 1 mM. Samples were mixed by pipetting and incubated 30 min prior to centrifugation. The samples were centrifuged for 60 min using a Beckman J2–21 ultracentrifuge equipped with a Beckman JA-21 rotor. The centrifuge was run at 50,000 G using compressed nitrogen. After the completion of each run, the supernatant was immediately removed and its absorption spectrum measured with a SOLAR CM 2203 spectrofluorometer. Bound (pelleted) protein was calculated as total minus free.

### 2.4. Estimation of the CytC Membrane-Permeabilization Activity

The activity of CytC mutant forms to induce pores in CL-containing membranes was estimated by the calcein leakage from CL-containing liposomes [25]. Calcein-loaded liposomes were prepared by evaporation under a stream of nitrogen of a 2% solution of a mixture of lipids in chloroform (4 mg soybean phosphatidylcholine (Sigma, Type II-S) and 1 mg bovine heart CL) followed by hydration with a buffer solution containing 50 mM calcein, 50 mM KCl, 5 mM Tris, and 5 mM MES, pH 7.4. The mixture was vortexed, passed through several cycles of freezing and thawing, and extruded through 0.1 mm pore size Nucleopore polycarbonate membranes using an Avanti Mini-Extruder. The unbound calcein was then removed by passage through a Sephadex G-50 coarse column with a buffer solution containing 100 mM KCl, 10 mM Tris, and 10 mM MES, pH 7.4. The fluorescence of the calcein-loaded liposomes was monitored at 520 nm (excitation at 490 nm) with a Panorama Fluorat 02 spectrofluorimeter (Lumex, Russia). The measurement solution contained 180 mM sucrose, 10 mM KCl, 10 mM Tris, 10 mM MES, 1 mM EDTA, pH 7.4; the lipid concentration was 10 mg/mL, CytC—1 mM; H_2_O_2_—1 mM. The extent of calcein efflux was calculated as (F_t_ − F_0_)/(F_100_ − F_0_), where F_0_ and F_t_ represent the initial fluorescence intensity and the fluorescence intensity at the time t, respectively, and F_100_ is the fluorescence intensity after the complete disruption of liposomes by the addition of the detergent Triton-X100 (final concentration, 0.1% *w*/*w*).

### 2.5. Peroxidase Activity Assay by Luminol Chemiluminescence

Peroxidase activity was assessed by chemiluminescence response [7,12,26] arising from the H_2_O_2_-induced oxidation of luminol in the presence of CytC. The assay was performed in sucrose buffer (200 mM sucrose, 10 mM Tris, 10 mM MES, pH 7.4) by adding 10 μM luminol and 25 μM H_2_O_2_ to a 0.5 μM solution of CytC or its mutant variants (lipid concentration, 10 µg/mL). Chemiluminescence was detected with a Lum-100 luminometer (DISoft LLC, Moscow, Russia), as described in Ref. [27].

### 2.6. Resonance Raman and Surface-Enhanced Raman Spectroscopy of CytC

The RRS and SERS spectra of wild-type and mutant cytochrome *c* molecules were recorded using confocal Raman spectrometer NTEGRA Spectra (NT-MDT, Zelenograd, Russia) with the 532 nm laser illumination and coupled to the inverted Olympus microscope, objective ×20 NA 0.45. The laser power for the registration spot with the diameter of appr. 800 nm was 3 and 0.3 mW under RRS and SERS spectra recording, respectively. The spectrum accumulation time was 20 s. All measurements were performed in 10 mM NaPi buffer, pH 7.0, 22 °C. Silver (Ag) nanostructured surfaces (AgNSSs) were prepared as described in [28] by means of Ag reduction in the silver ammonia complex, ultrasonically sprayed onto the coverslip surface and heated up to 340 °C for an hour with 5 min breaks every 3–4 min. The obtained AgNSSs were stored in the dark at room temperature and used in SERS experiments 2 weeks after synthesis. In order to record SERS spectra, the small volume (100 μL) of 10^−5^ M CytC was placed on a glass Petri dish and covered by AgNSS oriented with its nanostructured surface towards the microscope objective. The SERS spectra were recorded 1 min after the AgNSS placement into the CytC solution. In RRS experiments, we used solutions of reduced WT and mutant CytC at the concentration of 10^−5^ M without the AgNSS surface. The reduction of WT and mutant forms of CytC was performed with SDT. A small amount of SDT powder was added into the experimental probe with CytC 2–3 min before spectrum recording. In all cases, the number of independent measurements was 3–4.

### 2.7. Liposome Preparation

CL-containing liposomes were prepared, as described in the previous section, and incubated with the solution of oxidized CytC WT or its mutant forms for 1 h. After incubation, we centrifugated the mixture of liposomes with CytC and removed the supernatant with unbound CytC. Liposomes with the membrane-bound CytC were placed on the AgNSS, and SERS spectra were recorded with 532 nm laser excitation, laser power of 0.3 mW per 800 nm registration spot. The SERS spectrum accumulation time was 20 s.

## 3. Results and Discussion

### 3.1. CytC Binding to Cardiolipin-Containing Liposomes and the Peroxidative Permeabilization of Liposomes Induced by CytC Mutants

The ability of CytC mutant forms to bind to CL in liposomes was investigated using a spectrophotometric method. Figure 1A shows the absorbance spectra of WT CytC and its mutants with substitutions in the non-ordered red Ω-loop 70–85. Dashed curves show the spectra of supernatants obtained after the centrifugation of the corresponding mutant proteins incubated with CL-containing liposomes. As it is seen from the histogram in Figure 1B, all the CytC variants exhibited binding to the CL-containing liposomes, comparable to WT CytC binding. The exception was variant M3, the binding efficiency of which decreased by more than two times.

Figure 2A shows the time courses of calcein leakage from liposomes induced by the addition of horse heart WT CytC or its mutants in combination with H_2_O_2_, as monitored by an increase in calcein fluorescence. At the end of each calcein leakage trace, 0.1% Triton X-100 was added, and the level of calcein fluorescence recorded after the addition of Triton X-100 was used as 100% leakage. Figure 2B (black bars) displays the maximum rates of the liposome leakage (maximum slopes of the time-course curves) observed after the addition of each protein. It is seen that CytC mutants 8Mut, M1, and M2 induced peroxidative liposome leakage, with maximum rates comparable to that for WT CytC. A statistically valid decrease in the maximum rate of the CytC/H_2_O_2_-induced calcein leakage was found only in the case of the M3 mutant. However, this variant exhibited reduced CL-binding capacity (Figure 1B). 

A strikingly different behavior was demonstrated by the M4 mutant variant with the P76I/G77L/I81L/F82L substitutions: the maximum rate of the M4-CytC/H_2_O_2_-induced calcein leakage increased by more than five times compared to that of the WT-CytC/H_2_O_2_ complex.

Figure 3A shows the time courses of the intensity of luminol chemiluminescence induced by Cyt*С*/H_2_O_2_ in the presence of CL-containing liposomes. It is seen that the peroxidase-like activity of all mutant forms of CytC in the presence of CL-containing liposomes increased three to five times as compared to the activity of WT CytC, with the maximum (more than five times compared to WT) increase observed in variant M4 with P76I/G77L/I81L/F82L substitutions (Figure 3B). Remarkably, only in the presence of this CytC variant, the extraordinarily high rate of calcein release from liposomes was observed (Figure 2B). On the other hand, in contrast to a more than three-fold increase in the peroxidase-like activity of mutant forms 8Mut and M2 compared to WT CytC, no enhancement of the calcein release from liposomes in the presence of these mutant forms was observed (Figure 3B). Therefore, the data obtained do not enable us to assert the existence of a correlation between an increase in the peroxidase-like activity of CytC mutants and an increase in lipid membrane permeabilization induced by CytC mutants in the presence of H_2_O_2_.

Thus, the most striking result found here consists in dramatic stimulation of the peroxidative membrane-permeabilizing activity of CytC in the mutant variant P76I/G77L/I81L/F82L (M4). Based on recent findings [29,30] that the I81A and I81N mutations in human CytC exhibited a substantial enhancement in peroxidase activity, particularly below pH 7, it can be suggested that Ile81 plays a key role in regulating peroxidase and permeabilizing activity of CytC. Besides, the mutation at Pro76 was found to bring about a pronounced increase in peroxidase activity of human CytC [31]. Of note, the mutation of Phe82 was also shown to cause the destabilization of human CytC [32].

The obtained effects of the mutations on CytC-induced permeabilization of liposomes can be related to changes in the heme conformation of mutant proteins compared to the WT CytC heme. For this reason, we further investigated the conformational features of the hemoporphyrins of two CytC variants, 8Mut with substitutions of Lys residues from the universal binding site and M4 with four substitutions in the disordered Ω-loop 70–85, by the methods of resonance Raman (RRS) and surface-enhanced Raman spectroscopy (SERS) methods.

### 3.2. Conformation of Heme and Its Local Protein Environment in Cytochrome c Are Affected by the Mutations in the Ω-Loop and in the Universal Binding Site

We studied the conformations of heme in WT and mutant forms of reduced and oxidized CytC molecules in the phosphate buffer solution and in oxidized CytC bound to the outer surface of CL-containing liposomes. 

#### 3.2.1. Conformation of Heme in Unbound Reduced and Oxidized Cytochrome *c* Molecules 

Resonance Raman spectra of isolated reduced CytC molecules demonstrate a set of intensive peaks with maximum positions at 752, 1314, and 1585 cm^−1^, corresponding to methine bridges vibrations (CaCm bonds), and at 1130 cm^−1^ (vibrations of heme-CH_3_ groups) (Figure 4A,B) [16,28,33,34,35]. Further, the resonance Raman spectra of CytC possess less intensive peaks in the low-frequency spectrum range (below 750 cm^−1^) and around 1350–1400 cm^−1^ (Figure 4A). The peak at 571 cm^−1^, is a signature of the ruffled heme conformation and its relative intensity grows upon the increase in the probability of this conformation [16,33]. The peak with the maximum position at 1365 cm^−1^ corresponds to symmetric pyrrol half-ring vibrations in the heme of reduced *c*-type cytochromes and is not sensitive to the heme deformation. Previously, Sun and colleagues demonstrated that the ratio of peak intensities at 571 and 1365 cm^−1^ linearly increases with the shift of the Fe atom from the heme plane [33]. Peaks with maxima positions at 750, 1313, and 1585 cm^−1^ correspond to vibrations of all heme bonds and are considered to be partly sensitive to a change in the probabilities of planar/ruffled heme conformations. A peak with the maximum position at 1130 cm^−1^ corresponds to vibrations of CH_3_-side radicals in the heme. The relative intensity of this peak increases with a decrease in the local rigidity of the protein surrounding of the heme since the vibrational movements of atoms in CH3-radical become easier. Since the intensity of the peak at 1365 cm^−1^ does not depend on the heme conformation, it is usually used for the normalization of other peaks. So, we used the ratio of peak intensities I_571_/I_1365_ as the estimation of the probability of the ruffled heme conformation and the ratio I_1130_/I_1365_ as the estimation of the vibration probability of the CH_3_-radical in the CytC heme. The decrease in the rigidity of the protein part in the heme surrounding allows freer movements of the heme in the CytC cleft, resulting in the ease of CH_3_-radicals’ vibrations and in the increase in the I_1130_/I_1365_ ratio (Figure 4C,D). 

According to our data (Figure 4C,D), the reduced CytC with four site-directed mutations (M4) had a lower probability of the deformed ruffled heme conformation (corresponding to an increased probability of the planar heme conformation and to the positioning of Fe atom in the heme plane) than the WT and 8Mut cytochromes. For the M4 mutant, we also found changes in the probability of the heme CH_3_-bond vibrations, demonstrating the decrease in the rigidity of the CytC protein part in the local surrounding of the heme (Figure 4C,D). We did not observe changes in the planarity of the heme or in the probability of heme CH_3_-group vibrations in the CytC mutant with eight site-directed mutations (Figure 4C,D). 

Heme conformations in oxidized WT and mutant forms of CytC in the buffer and in the membrane-bound state were studied by means of surface-enhanced Raman spectroscopy because oxidized CytC mutants possess low Raman intensity, undetectable without AgNSSs. Previously, with a wide range of preparations, we demonstrated the stability of the AgNSSs in physiological buffers and the absence of their effect on the lipids and proteins of different membranes, purified CytC in the buffer, the morphology of erythrocytes, and the functional activity of mitochondria [16,28,36,37]. Thus, AgNSSs do not influence WT and mutant CytC molecules in buffer and can be used to study oxidized CytC in the buffer or under interaction with CL-containing liposomes. The SERS spectra of oxidized WT and its mutant forms in buffer solution have many pronounced peaks with intensity maxima at the following positions: 1130, 1168, 1314, 1375, 1570, and 1638 cm^−1^. Peaks at 1314 and 1570 cm^−1^ correspond to the vibrations of all heme bonds, peaks at 1168 and 1375 cm^−1^ correspond to asymmetric and symmetric pyrrol half-ring vibrations that do not depend on the heme planarity, the peak at 1638 cm^−1^ corresponds to the methine bridges’ vibrations in planar heme, and the peak at 1130 cm^−1^ corresponds to the vibrations of heme CH_3_-side radicals (Figure 5A). The SERS spectra of oxidized CytC molecules bound to CL-containing liposomes had a similar structure to the SERS spectra of unbound CytC in the buffer solution (spectra are not shown). We used the ratio of peak intensities I_1638_/I_1375_ as the estimation of the probability of the planar heme conformation, the ratio I_1130_/I_1375_ as the estimation of the probability of CH_3_-group vibrations, and the rigidity of the heme local surrounding and the ratio I_1168_/I_1375_ as the estimation of the probability of asymmetric pyrrol ring vibrations vs symmetric vibrations.

We observed an increase in the probability of the planar heme conformation (a decrease in the probability of the ruffled heme conformation) in CytC mutants with four and eight mutations compared to WT CytC (Figure 5B), which indicated the location of the Fe atom closer to the heme plane in the M4 and 8Mut proteins than in WT cytochrome *c*. The binding of CytC to CL in liposomes differently affects the heme planarity in various mutants: WT CytC binding to liposomes caused an increase in the probability of heme plane conformation, whereas, in 8Mut CytC, the probability of the plane conformation was decreased with respect to unbound 8Mut CytC (Figure 5B, white and light violet bars). The binding of M4 CytC to liposomes did not affect the planarity of the heme (Figure 5B, grey and bright violet bars). Thus, even upon the binding of 8Mut CytC to the liposomal membrane, its heme had a higher probability of plane conformation compared to WT and M4 CytC. We also found a decrease in the probability of the heme CH_3_-group vibrations and the decrease in the probability of asymmetric pyrrol ring vibrations in the mutants with four and eight site-directed mutations compared to WT CytC in both states of CytC: unbound molecules in buffer and in the membrane-bound CytC molecules in liposomes (Figure 5C,D). This result indicates higher rigidity in the protein surrounding the heme of CytC with four and eight site-directed mutations (Figure 5). Such an increase in the rigidity of the heme surrounding oxidized M4 and 8Mut CytC mutants may be responsible for their decreased ability to accept an electron from the cytochrome *c_1_* in complex III [17], since for the electron tunneling between cytochrome c*_1_* and CytC, the heme *c* should obtain an optimal orientation towards heme *c_1_* that can be obstructed in the rigid heme surrounding the studied CytC mutants. 

#### 3.2.2. Ordering of Lipids in CL-Containing Liposomes upon CytC Binding

To evaluate the effect of CytC binding to CL-containing liposomes on the membrane lipids’ ordering, we recorded SERS spectra in the high-frequency range (2600–3100 cm^−1^) containing peaks attributed to the bond vibrations of liposome lipids (Figure 6). Table 1 shows the positions of peak maxima and their assignment to certain bond vibrations. Table 2 demonstrates the SERS spectra parameters that were used to analyze the conformation of lipids in Cl-containing liposomes with and without bound CytC.

It is seen that in the case of CytC with four site-directed mutations, the ratio I_2935_/I_2860_ is significantly increased compared to WT CytC, and there is a tendency to down-shift the peak corresponding to the symmetric vibrations of C-H bonds in CH_3_-groups. For both CytC mutant forms, we observed a significant increase in the ratio I_2935_/I_2905_ compared to WT CytC. These results indicate that the interactions of CytC mutants with CL in liposomes cause a decrease in the ordering of lipid tails in liposomes. For both mutant cytochrome forms and WT, we also analyzed the spectral region 2860–2890 cm^−1^ that indicates, to some extent, acyl chain lateral packing density. It is known that for pure lipid membranes or lipid structures, the acyl chain lateral packing density correlates with the ratio I_2883_/I_2847_. When the packing density is high, as in the case of ordered acyl chains, dipolar coupling between methylene C−H vibrations in adjacent chains leads to an increase in the intensity of the CH_2_ symmetric stretching mode and to a decrease in the I_2883_/I_2847_ ratio. In the spectra of cytochrome *c* bound to CL-containing liposomes, the protein backbone C−H stretching modes overlap with the antisymmetric stretching mode (2883 cm^−1^) of the phospholipid acyl chains, making the I_2883_/I_2847_ ratio a poor indicator of bilayer structure. However, in the different SERS spectra of WT CytC-liposomes and mutant CytC-liposomes, a reduction in the C−H symmetric stretching mode was observed, which indicated a decrease in acyl chain methylene C−H vibrational coupling arising from disordered acyl chains. All these data demonstrate that the binding of both CytC mutants to CL-containing liposomes, compared to WT CytC binding, caused the disordering of the acyl chains of Cl and PC.

It should be noted that the interaction of WT CytC with Cl-containing liposomes also caused an increase in the I_2935_/I_2860_ ratio compared to pure liposomes, which indicated the disordering of lipid chains. 

## 4. Conclusions

In summary, we have found that site-directed mutations P76I/G77L/I81L/F82L in the Ω-loop of CytC caused multiple changes in the conformation of heme and the protein part of oxidized CytC. In particular, we observed that in this mutant, in the oxidized state, the probability of the plane heme conformation was higher compared to WT CytC, but the mobility of heme CH_3_-groups was lower, thereby indicating a higher rigidity of the protein heme surrounding the mutants compared to WT CytC. Such an increase in the rigidity of the heme surrounding may obstruct the optimal orientation of the mutant CytC heme towards the heme of cytochrome c*_1_*, deteriorating the electron transfer from the complex III to CytC. A similar change in the heme conformation and its surrounding was observed for the CytC mutant with eight site-directed mutations in the universal binding site (K8E/K27E/K72E/K86E/K87E/E62K/E69K/E90K). The binding of oxidized WT and mutant CytC molecules to CL-containing liposomes caused the disordering of lipid chains and the deterioration of CL and PC chain interaction. This effect was more pronounced for the mutant with four site-specific mutations in the Ω-loop (P76I/G77L/I81L/F82L) compared to the mutant K8E/K27E/K72E/K86E/K87E/E62K/E69K/E90K (8Mut) or WT CytC. In line with these observations, the strong stimulation of the peroxidative membrane permeabilizing activity of CytC in the mutant variant P76I/G77L/I81L/F82L was found. Of note, the CL-binding ability was similar for WT CytC and both its mutants. The obtained results demonstrate that the Ω-loop is important for the interaction with the membrane phospholipids and that the increase in the rigidity in this loop leads to the increased permeabilizing activity of cytochrome *c*.

## Figures and Tables

**Figure 1 biomolecules-12-00665-f001:**
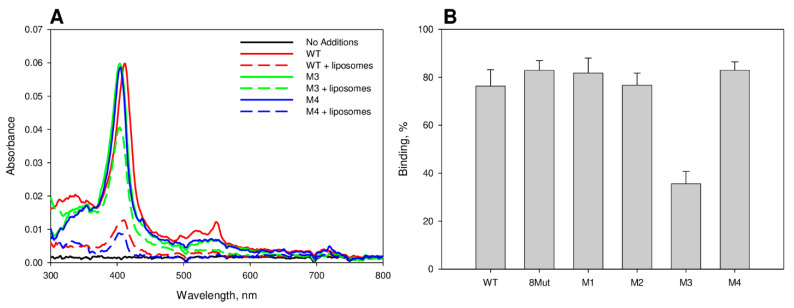
Absorbance spectra (**A**) of supernatants obtained after centrifugation of WT CytC and its mutants with various substitutions in the absence (solid lines) and in the presence (dashed lines) of CL-containing liposomes, and a histogram (**B**) showing the extent of the binding of mutant forms K8E/K27E/K72E/K86E/K87E/E62K/E69K/E90K (8Mut), T78N/K79Y/M80I/I81M/F82N (M1), T78S/K79P (M2), I81Y/A83Y/G84N (M3), and P76I/G77L/I81L/F82L (M4) to the liposomes. Results are expressed as mean ± SE of at least three experiments.

**Figure 2 biomolecules-12-00665-f002:**
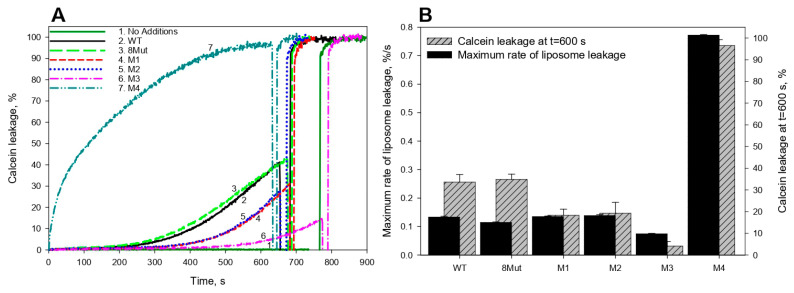
(**A**) Time courses of calcein leakage from liposomes induced by the addition of WT CytC or its mutants with H_2_O_2_ and (**B**) maximum rates of the liposome leakage measured after the addition of the mutant proteins (8Mut-K8E/K27E/K72E/K86E/K87E/E62K/E69K/E90K, M1-T78N/K79Y/M80I/I81M/F82N, M2-T78S/K79P, M3-I81Y/A83Y/G84N, M4-P76I/G77L/I81L/F82L). Results are expressed as mean ± SE of at least three experiments. At the end of each leakage trace, 0.1% Triton X-100 was added, and the level of calcein fluorescence recorded after the addition of Triton X-100 was used as 100% leakage.

**Figure 3 biomolecules-12-00665-f003:**
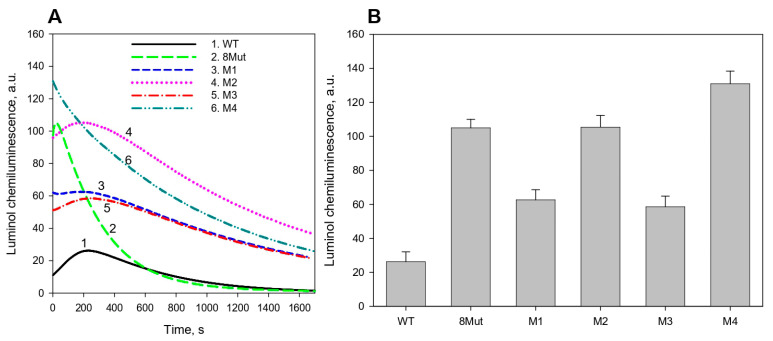
(**A**) Time courses of the intensity of luminol chemiluminescence induced by CytC/H_2_O_2_ in the presence of CL-containing liposomes and (**B**) maximum values of luminol chemiluminescence after the addition of H_2_O_2_ and CytC mutant forms (8Mut-K8E/K27E/K72E/K86E/K87E/E62K/E69K/E90K, M1-T78N/K79Y/M80I/I81M/F82N, M2-T78S/K79P, M3-I81Y/A83Y/G84N, M4-P76I/G77L/I81L/F82L). Results are expressed as mean ± SE of at least three experiments.

**Figure 4 biomolecules-12-00665-f004:**
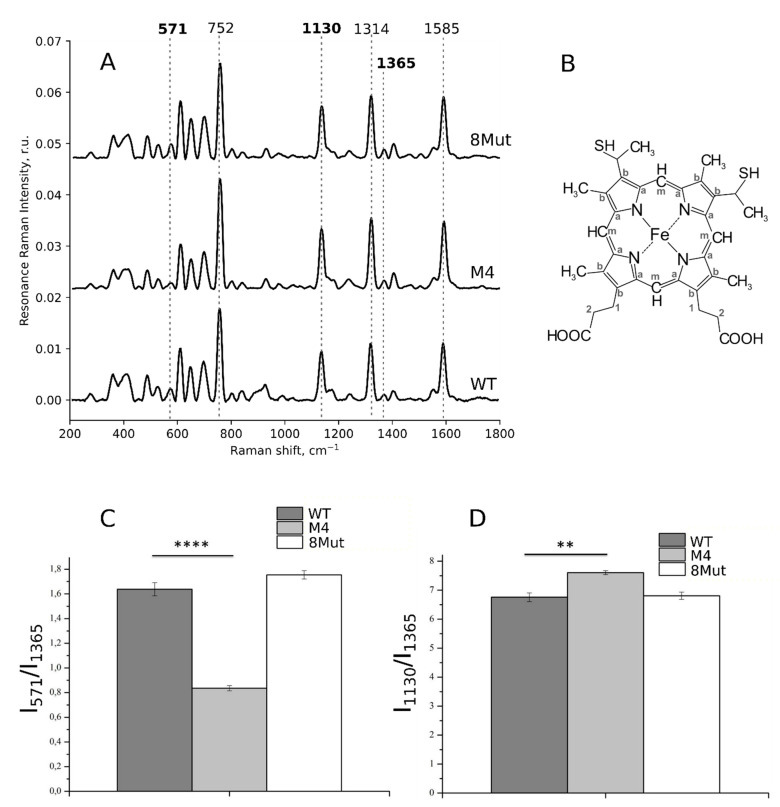
(**A**) Resonance Raman spectra of reduced WT CytC and its mutant forms in PBS buffer solution (10^−5^ M) with four and eight site-directed mutations (M4 and 8Mut, respectively). The reduction of CytC was performed by adding SDT powder to the experimental chamber 2–3 min before the spectrum registration. For a clearer representation, spectra are normalized on the sum of all spectrum intensities and shifted vertically. Dotted lines show the maximum positions of the most intensive peaks and of a peak corresponding to ruffled heme conformation (at 571 cm^−1^) or a planarity-insensitive peak (at 1365 cm^−1^). Numbers above peaks show maximum peak positions. Bold numbers demonstrate peaks used in the study. (**B**) The structural formula of heme *c*. (**C**,**D**) Ratios of peak intensities calculated from the resonance Raman spectra of reduced WT CytC and its mutants with four and eight site-directed mutations (dark grey, light grey, and white boxes, respectively). The I_571_/I_1365_ ratio corresponds to the probability of the ruffled heme conformation and the I_1130_/I_1365_ ratio corresponds to the probability of vibrations of the heme CH_3_-side radicals (right figure). Data are presented as mean values and the error of the mean, *n* = 7. Statistics were calculated with ANOVA (GraphPad Prism 8.0). ** *p* < 0.01, **** *p* < 0.0001.

**Figure 5 biomolecules-12-00665-f005:**
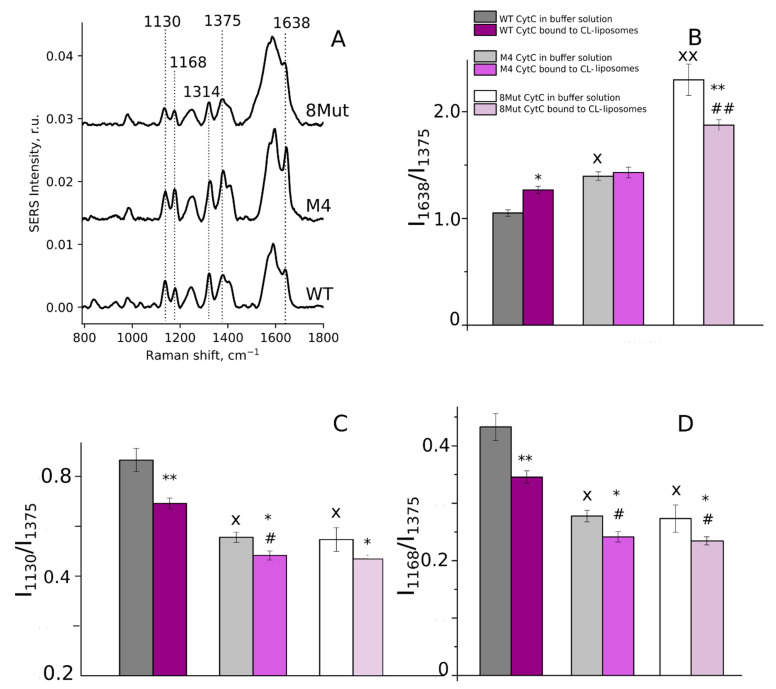
(**A**) SERS spectra of oxidized WT, M4, and 8Mut cytochrome *c* (1 mM) in the phosphate buffer. For clearer presentation, spectra are normalized to the sum of all spectrum intensities and shifted vertically. Dotted lines show the maximum positions of the most intensive peaks used for the heme conformation analysis. (**B**–**D**) Ratios of peak intensities calculated from the SERS spectra of WT CytC and its mutants with four and eight site-directed mutations in buffer solution (dark grey, light grey, and white boxes, respectively) and in the membrane-bound state in CL-containing liposomes (dark violet, bright violet, and light violet boxes, respectively). The I_1638_/I_1375_ ratio corresponds to the probability of the plane heme conformation (**B**), the I_1130_/I_1375_ ratio corresponds to the probability of the heme CH_3_-group vibrations (**C**), and the I_1168_/I_1375_ ratio corresponds to the probability of asymmetric pyrrol ring vibrations vs their symmetric vibration. Data are presented as mean values and the error of the mean, *n* = 7. Statistics were evaluated with ANOVA (GraphPad Prism 8.0). * *p* < 0.05, ** *p* < 0.01 for the specified ratios calculated for a certain CytC form bound to CL-containing liposomes compared to this CytC form in the buffer. # *p* < 0.01, ## *p* < 0.001 for the specified ratios for M4 and 8Mut in liposome-bound state compared to WT CytC in the liposome-bound state. × *p* < 0.01, ×× *p* < 0.001 for the specified ratios for M4 and 8Mut in the buffer compared to WT CytC in the buffer.

**Figure 6 biomolecules-12-00665-f006:**
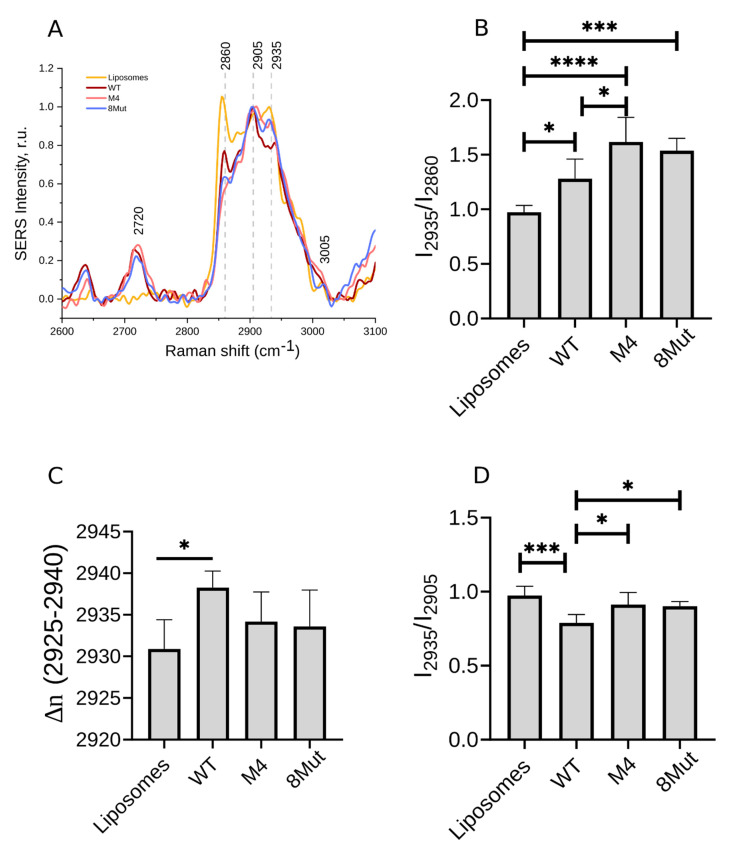
(**A**) Typical SERS spectra of control liposomes (PC:CL = 4:1) and liposomes with added cytochromes, normalized to the intensity of the peak 2905 cm^−1^. (**B**) The relative input of the symmetric vibrations of C−H bonds in CH_3_ groups compared to the symmetric vibrations of −CH_2_− backbone bonds in the CytC protein part and in the acyl chains of liposomal lipids. (**C**) Maximum position of the peak corresponding to the symmetric vibrations of C−H bonds in side-chain CH_3_-groups. (**D**) The relative input of the symmetric vibrations of C−H bonds in −CH3 radicals compared to the asymmetric vibrations of −CH2− backbone bonds. Data are presented as mean values and the error of the mean, *n* = 7. Statistics were calculated with ANOVA (GraphPad Prism 8.0). * *p* < 0.05, *** *p* < 0.001, **** *p* < 0.0001.

**Table 1 biomolecules-12-00665-t001:** Raman peak assignment (done in accordance with [38,39]).

Position of a Peak Maximum in SERS Spectra CL-Containing Liposomes with Bound CytC	Raman Peak Position in Spectra of CL-Containing Liposomes without CytC	Description	Comments
2627	-	ν(C–H)	
2720	-	ν(C–H)	
2860	2854	ν_s_(-CH_2_-)	
2905	2906	ν_as_(-CH_2_-)	
2925–2940	2930	ν_s_(CH_3_)	
3005	3014	н(=CH-)	Correlates with the number of double C=C bonds in CL [39] and is affected by CytC-specific interaction with CL

**Table 2 biomolecules-12-00665-t002:** Peak ratios used for the analysis of CytC interaction with the CL of liposomes.

Parameter, Calculated from Raman Spectra	Meaning	Comments
I_2935_/I_2860_	The relative input of symmetric vibrations of C-H bonds in amino acids’ CH_3_-groups compared to the symmetric vibrations of -CH_2_- backbone bonds	The ratio increase corresponds to the easier -CH_3_ bond vibrations due to the decrease in the ordering of the membrane lipid tails and due to the appearance of disturbances in the membrane lipid phase
I_2935_/I_2905_	The relative input of the symmetric vibrations of C-H bonds in amino acids’ CH_3_-groups compared to the asymmetric vibrations of -CH_2_- backbone bonds	The ratio gives similar information as the ratio I_2935_/I_2860_. The disordering of lipid tails affects this ratio to a greater degree than the ratio I_2935_/I_2860_
ν(2925–2940)	The peak corresponds to the symmetric vibrations of C-H bonds in amino acids’ CH_3_-groups	Peak maximum shifts under changes in the lipid ordering and membrane microfluidity in the CytC surrounding

## Data Availability

All raw data are available from corresponding authors under reasonable request.

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
