# Peer review of "Multiple Mutations in the Non-Ordered Red Ω-Loop Enhance the Membrane-Permeabilizing and Peroxidase-like Activity of Cytochrome c"

_biomolecules, 2022, doi:10.3390/biom12050665_

Round 1
Reviewer 1 Report
The subject of the paper is appropriate for the publication in the journal Biomolecules. The submitted paper is an article dealing mainly with an influence of mutations in Ω-loop of cytochrome c (cyt c) on the peroxidase-like activity and membrane permeabilizing properties of this enzyme. The authors have found that the four site-directed mutations in Ω-loop increase peroxidase like activity, change the ability of cyt c to permeabilize membrane containing cardiolipin and leads to increase of the probability of the plane heme conformation in the protein. Generally, the manuscript is written well with clear conclusions. However, before I can recommend this paper to be published, the authors should answer and clarify the following points:
- The word “dramatically” in the title of the manuscript is a relatively strong expression for the observed results. I recommend to omit this word from the title.
- The abstract should be shortened. In the current version is too extensive.
- In the Introduction part is written that “the peroxidase activity of cyt c is induced by the formation of a complex between cyt and cardiolipin” (p.2, l. 59-60). Is it possible that cyt c has a peroxidase like activity without binding to cardiolipin?
- In the SERS experiments, the gold nanoparticles were used as a substrate for Raman signal enhancement. Can the presence of these nanoparticles influence the structure, function and stability of cyt c?
- The Fig. 1A is not correct. It seems that several spectra on this Fig. presents negative values of the absorbance. Can you explain this fact?
- Fig. 2 shows time course of calcein leakage from liposomes induced by cyt c and the mutants of this protein. However, it is not clear why the signal corresponding to calcein leakage dramatically increase after ca. 600 sec in every sample. Also I would like to ask if the authors performed control experiments of the calcein leakage from liposome in the presence of cyt c (and the mutants), however, in the absence of H2O2.
- The up and down profile of the time course (1,3,4,5) of luminol chemiluminescence should be explained. The time courses 2 and 6 show only decrease of the chemiluminescence intensity of luminol.
- The tile of the paragraph 3.2 is too long, it should be shortened.
- It is not clear what does the expression “the probability of the heme CH3 group vibrations” (p. 9, l. 322, p. 10, l. 334, p. 1, l. 367) mean.
- The description of the axes (font size) on the graphs is not unified.
- The Table 2 is not necessary to be involved in the text.
- Only five citations from the last five years are involved in the List of references. I think the number of the citations of current publications in the topic should be increased.
Only minor grammatical and stylistic errors are present in the manuscript,
Recommendation: Before publication, the authors should address the above mentioned points.
Author Response
We thank the Reviewer for the careful reading of the manuscript and valuable comments. With Reviewer’s permission, we give answers to comments step by step.
- The word “dramatically” in the title of the manuscript is a relatively strong expression for the observed results. I recommend to omit this word from the title.
Answer. The word “dramatically” was omitted in the revised version.
- The abstract should be shortened. In the current version is too extensive.
Answer. The abstract was shortened.
- In the Introduction part is written that “the peroxidase activity of cyt c is induced by the formation of a complex between cyt and cardiolipin” (p.2, l. 59-60). Is it possible that cyt c has a peroxidase like activity without binding to cardiolipin?
Answer. Yes, it is known that cytochrome c has a weak peroxidase-like activity (R.E. Diederix et. al, 2002; A. Lawrence et. al, 2003). However, when cytochrome binds to cardiolipin, peroxidase activity dramatically (many times) increases (V.E. Kagan et. al, 2005; N.A. Belikova et.al, 2006; Y.A. Vladimirov et. al,2009). In addition to binding to cardiolipin, other factors lead to the same effect, for example, partial proteolysis, denaturation, dimerization, tyrosine nitration and others.
- In the SERS experiments, the gold nanoparticles were used as a substrate for Raman signal enhancement. Can the presence of these nanoparticles influence the structure, function and stability of cyt c?
Answer. In SERS experiments we used plasmonic silver nanostructures on the glass surface. These Ag nanostructured surfaces do not contain any by-products of their synthesis and are stable in various physiological buffers. Previously we used these Ag nanostructured surfaces with various preparations: isolated mitochondria (in physiological buffer), purified cytochrome C, erythrocytes, erythrocytic ghosts and liposomes and we demonstrated absence of nanostructures’ effect on the studied objects [36, 37]. One of various proofs of the absence of AgNSSs effect on the studied objects is the stability of recorded SERS spectra in time and the high reproducibility of SERS spectra measured from different regions of AgNSS or from different AgNSSs. Mitochondria placed on AgNSSs preserved their functional activity, erythrocytes maintain their discoidal shape demonstrating absence of the nanostructure’s influence on proteins and lipids of mitochondria and erythrocyte membranes. We added the description of AgNSS and of the absence of their influence on biological preparations into the manuscript (lines 390-403)
- The Fig. 1A is not correct. It seems that several spectra on this Fig. presents negative values of the absorbance. Can you explain this fact?
Answer. We agree with the reviewer that the quality of the spectra is low apparently due to low concentration of the protein and low resolution of the spectrophotometer. We reproduced these binding experiments using high resolution spectrophotometer (namely Specord 50, Jena, Germany) and found qualitatively similar binding results. However, the spectra quality has become better (new Figure 1a of the revised version).
- Fig. 2 shows time course of calcein leakage from liposomes induced by cyt c and the mutants of this protein. However, it is not clear why the signal corresponding to calcein leakage dramatically increase after ca. 600 sec in every sample. Also I would like to ask if the authors performed control experiments of the calcein leakage from liposome in the presence of cyt c (and the mutants), however, in the absence of H2O2.
Answer. We apologize for not describing the effect of 0.1 % Triton X-100 at the end of each calcein leakage trace, although we wrote about this in the Methods. The level of calcein fluorescence after the addition of Triton X-100 was used as 100 % leakage. We added the description in the text and the Figure legend.
We did not see any liposome leakage in the control experiments in the absence of H2O2
- The up and down profile of the time course (1,3,4,5) of luminol chemiluminescence should be explained. The time courses 2 and 6 show only decrease of the chemiluminescence intensity of luminol.
Answer. The measurements of peroxidase activity of cytochrome c via chemiluminescence in the presence of luminol have been introduced in the laboratories of Radi, Vladimirov, and Kagan. We cited these works in the Methods, subsection “Peroxidase activity assay by luminol chemiluminescence” (refs. 7, 12, 26, 27). It was shown, in particular, that “the kinetics of the chemiluminecence traces are consistent with an initial activation of cytochrome c by H2O2 to a catalytically more active species in which a high oxidation state of an oxo-heme complex mediates the oxidative reactions“ (ref. 26). Although the system was under study for more than 30 years, the kinetic scheme of the process still remains controversial. However, it was accepted that the amplitude of the chemiluminescence traces can be related directly to the rate of the peroxidase activity of cytochrome c (ref. 27).
- The tile of the paragraph 3.2 is too long, it should be shortened.
Answer. Done
- It is not clear what does the expression “the probability of the heme CH3 group vibrations” (p. 9, l. 322, p. 10, l. 334, p. 1, l. 367) mean.
Answer. The expression “the probability of the heme CH3-group vibrations” refers to the vibrational movement of the whole CH3-side radical in the heme relatively to the pyrrol rings in the heme. The increase in the probability of CH3-side radicals’ vibrations means more frequent periodical change in the length of the bond between heme and C atom in CH3-group. We modified the description of the peak at 1130cm-1 and of the ratio I1130/I1365 (lines 334-337, 339-343, 401).
- The description of the axes (font size) on the graphs is not unified.
Answer. Corrected
- The Table 2 is not necessary to be involved in the text.
Answer. We have shortened significantly the Table 2. However, we would like to keep it in the manuscript, since the description of the proposed peak ratios in the main text will complicate the reading of the results.
- Only five citations from the last five years are involved in the List of references. I think the number of the citations of current publications in the topic should be increased.
Answer. We added an abstract discussing the liposome leakage data on page 7 with 4 additional references on relevant current publications.
Only minor grammatical and stylistic errors are present in the manuscript,
Answer. Corrected
Recommendation: Before publication, the authors should address the above mentioned points.

Reviewer 2 Report
The manuscript titled “Multiple mutations in the non-ordered red Ω-loop dramatically increase membrane-permeabilizing and peroxidase-like activity of cytochrome c” by R.V. Chertkova et al. deals with the interesting and very important biological event, apoptosis, particularly with the cytochrome c -dependent apoptotic pathway and role of interaction of cytochrome c with cardiolipin in apoptosis.
In previous published works, the authors constructed a number of cytochrome c (CytC) mutants. In the submitted manuscript the ability of the recently constructed mutants to bind to cardiolipin, and their membrane-permeabilization and peroxidase activities were investigated.
Overall, the manuscript is well written and relevant for the field of the Journal.
The selected methods are adequate for solving the tasks.
To improve the manuscript, I have a few suggestions/remarks.
Lines 72-75. The authors formulated a weak statement regarding the importance of work. The authors stated “…it seems relevant…” and “…would probably contribute...”. It is recommended to improve this statement and make it stronger.
Line 109. Cardiolipin. Was cardiolipin purchased in chloroform or powder? Please, clarify.
Line 109. The abbreviation SDT should be in round brackets.
Line 200. The abbreviation for sodium dithionite (SDT) was already introduced (Line 109).
Figure 1, Panel A. It was stated, that “Fig. 1A shows absorbance spectra of WT CytC and its mutants with substitutions in the universal binding site (8Mut) and in the non-ordered red Ω-loop 70-85 (M1–M4)” However, only WT, M3, and M4 spectra are displayed. Please, clarify.
Figure 4. If the Resonance Raman spectra are taken using reduced WT CytC as stated in Line 304 then it should be mentioned in Figure Legend (Resonance Raman spectra of reduced WT CytC …).
Please, check the spelling. For example, the word “permebializing” or “permebialization”.
Author Response
We thank the Reviewer for the careful reading of the manuscript and valuable comments. With Reviewer’s permission, we give answers to comments step by step.
- Lines 72-75. The authors formulated a weak statement regarding the importance of work. The authors stated “…it seems relevant…” and “…would probably contribute...”. It is recommended to improve this statement and make it stronger.
Answer. The sentence is modified to the following: “In this regard, it seems important to study the role of individual functionally significant amino acid residues or sequences of CytC in the interaction of CytC with CL-containing membranes. The data obtained will make a certain contribution to the determination of the molecular mechanisms of CytC interaction with mitochondrial membranes during its translocation into the cytosol, expanding our understanding of the role of conformational changes in CytC during initial stages of apoptosis.”
- Line 109. Cardiolipin. Was cardiolipin purchased in chloroform or powder? Please, clarify.
Answer. Cardiolipin was purchased in powder, included into the manuscript.
- Line 109. The abbreviation SDT should be in round brackets.
Answer. Corrected
- Line 200. The abbreviation for sodium dithionite (SDT) was already introduced (Line 109).
Answer. Corrected
- Figure 1, Panel A. It was stated, that “Fig. 1A shows absorbance spectra of WT CytC and its mutants with substitutions in the universal binding site (8Mut) and in the non-ordered red Ω-loop 70-85 (M1–M4)” However, only WT, M3, and M4 spectra are displayed. Please, clarify.
Answer. We corrected this inconsistency. We presented the spectra only of WT, M3, and M4 because other spectra are very similar and they overlap each other.
- Figure 4. If the Resonance Raman spectra are taken using reduced WT CytC as stated in Line 304 then it should be mentioned in Figure Legend (Resonance Raman spectra of reduced WT CytC …).
Answer. Corrected
- Please, check the spelling. For example, the word “permebializing” or “permebialization”.
Answer. The spelling was checked and corrected.

Reviewer 3 Report
Please see attached review

Author Response
We thank the Reviewer for the careful reading of the manuscript and valuable comments. With Reviewer’s permission, we give answers to comments step by step.
- Figure 1:
- The text indicates that Figure 1A should show UVVis spectra of WT, M1, M2, M3, M4 and 8Mut with and without CL. However, Figure 1A shows only spectra of WT, M3 and M4 with and without CL.
Answer. We corrected this inconsistency. As we wrote above, all spectra are very similar and overlap each other.
- The presented data in Figure 1A seem incomplete. The whole range of 250 – 800 nm should be shown, so the changes associated with mutation and CL addition in the Soret and Q bands can be observed.
Answer. We carried out new measurements and presented full spectra in the revised version.
- The baselines of all shown spectra (except for the spectrum of WT) fall below 0 absorption; these measurements should be repeated with a corrected baseline.
Answer. We carried out new measurements and presented more accurate spectra in the revised version.
- There is no discussion of the data shown in Figure 1.
Answer. The data shown in Fig.1 show that the binding of all mutants studied is very similar. Only M3 mutant exhibits a bit less binding. This conclusion is important for our study of the mechanism of the permeabilizing ability of these mutants.
- What are the conclusions of studies shown in Figure 2? It would be helpful if the authors
provided a more elaborate discussion than the 237-240 lines.
Answer. The data presented in Fig.2 show that the permeabilizing activity of M4 mutant exceeds the activity of the others by many fold. This result is central in the manuscript and is discussed later on.
- The first sentence in the 3.2.1 paragraph (lines 282 – 286) should have a reference.
Answer. Corresponding references are included
- The experimental part lacks a description of how the reduced form of CytC was generated;
please add appropriate procedures.
Answer. The details about CytC reduction with sodium dithionite is added to the section 2.6 of “Materials and Methods”, lines 202-203 and into the legend of the Figure 4 (lines 326-329).

Round 2
Reviewer 1 Report
The authors have satisfactorily answered all questions and comments. The quality of the manuscript has been improved.Although there are still present some few grammatical, stylistic, and formal errors in the manuscript (I hope that final critical reading will eliminate them), I can fully recommend the manuscript for the publication in journal Biomolecules